ecology

bats, Chiroptera, Costa Rica, mist nets

**Author for correspondence:**
Gloriana Chaverri
e-mail: gloriana.chaverri@ucr.ac.cr

# Comparing the efficiency of monofilament and regular nets for capturing bats

Silvia Chaves-Ramírez[1], Christian Castillo-Salazar[1], Mariela Sánchez-Chavarría[2], Hellen Solís-Hernández[2] and Gloriana Chaverri[2,3]

[1]Escuela de Biología, Universidad de Costa Rica, San Pedro, Costa Rica
[2]Sede del Sur, Universidad de Costa Rica, Golfito, Costa Rica
[3]Smithsonian Tropical Research Institute, Balboa, Ancón, Panamá

SC-R, 0000-0002-0377-2167; CC-S, 0000-0003-0468-983X;
GC, 0000-0002-1155-432X

Regular nylon or polyester mist nets used for capturing bats have several drawbacks, particularly that they are inefficient at sampling insectivorous species. One possible alternative is to use monofilament nets, whose netting is made of single strands of yarn instead of several as regular nets, making them less detectable. To date, only one study has quantified the differences in capture rates between monofilament and regular mist nets for the study of bats, yet surprisingly, its findings suggest that the latter are more efficient than the former. Here, we provide further evidence of the differences in sampling efficiency between these two nets. We captured 90 individuals and 14 species in regular nets and 125 individuals and 20 species in monofilament nets. The use of monofilament nets increased overall capture rates, particularly for insectivorous species. Species accumulation curves indicate that samples based on regular nets are significantly underestimating species diversity, most notably as these nets fail at sampling rare species. We show that incorporating monofilament nets into bat studies offers an opportunity to expand records of different guilds and rare bat species and to improve our understanding of poorly known bat assemblages while using a popular, relatively cheap and portable sampling method.

## 1. Introduction

In animal research studies, it is often necessary to capture the study organism, and the sample of individuals that are trapped should ideally be representative of the target species or assemblage. However, there are many problems associated with trapping methods that hamper the quality of data obtained, most notably

differences in capturing success among individuals or species. A well-known phenomenon, for example, is the difference in trap-shy or trap-happy individuals, or those that are either less or more prone to being recaptured, respectively, which creates serious sampling biases [1]. Also, some individuals within a species are easily trapped while others are not [2], and some sampling methods are consistently more effective than others, even within the same taxon [3,4]. These biases may significantly affect a study's results and their interpretation, which deteriorates the decisions involving the taxon under study or its habitats. For example, capture methods may bias estimates of density and population structure [2], population trends [5] and estimates of species richness and capture rates [6]. It is still possible to account for capture probabilities as a way to correct for some of these biases [2], yet this requires thorough knowledge of a trapping method's effect on all target individuals and species [5,7], data which are commonly not available for the majority of species. Therefore, selecting an appropriate and representative trapping method is critical for making the correct inferences and appropriate decisions concerning the focal organism.

Bats are the most widely distributed terrestrial mammals on Earth and constitute almost one-fifth of mammalian biodiversity [8]. However, their ability to fly and nocturnal habits make them a difficult group to study [9]; thus, efficient sampling methods are essential for their capture and identification. Despite their large diversity and wide distribution, information on many species is still deficient, especially in areas that harbour the greatest diversity [10]. A wide variety of methods exist for the study of bats, which differ in effectiveness and practicality depending on the goal of the study. Among the most commonly known methods for capturing bats are mist nets, harp traps, hand nets and direct captures at their roosts [11]. Indirect recording of species has increased with the use of camera traps, thermal cameras and acoustic recording equipment [12].

Bat capture methods, such as mist nets, are considered invasive as they increase the stress associated with the capture process and require substantial previous experience by the user, especially during the process of extracting individuals from the net [13,14]. However, mist nets are essential for collecting information on morphometrics, to create acoustic libraries, understand sensitivity to ectoparasites, gauge species diversity and estimate individual health, among others. An additional problem with the use of mist nets is that some species are very difficult to capture, either because they fly extremely high or because their sensory abilities allow them to detect and avoid the nets [15,16]. On the other hand, indirect methods for studying bats, such as acoustic monitoring, may be advantageous since they do not cause stress to the animals and may be able to record individuals that fly very high and/or are difficult to capture. Acoustic monitoring studies, however, depend on the availability of sufficiently large and representative acoustic libraries, which are not yet available for many sites and species [17]. In addition, the use of acoustic recording equipment imposes higher economic costs than other existing capture methods and does not provide data such as sex, age, body condition and other data that may be relevant to a large number of studies.

Despite the difficulties described above, mist nets remain an essential, practical and accessible method for capturing bats, facilitating research and monitoring of species worldwide. Some of the types of mist nets that have been developed, according to their netting, include polyester and nylon which have thicker threading, and ultrathin nets such as hair and monofilament nets. The latter types were developed as a novel tool for bat trapping, and these could have great potential to minimize some of the detectability limitations of other types of mist nets, while still benefiting from the already well-described advantages of this trapping method. Monofilament nets are designed to be less detectable by bats, as their netting is made of single strands of yarn, unlike regular (i.e. polyester and nylon) mist nets whose netting is created by twisting several individual strands, resulting in a thicker netting material. However, to date only one study has quantified their capture efficiency compared with regular mist nets [18]. The results of the study by Ferreira *et al*. [18] suggest that regular nets are more efficient than monofilament ones, which is surprising given that many species of bats are expected to detect larger objects, in this case a thicker netting material as that found in regular nets, more readily than smaller objects [19]. Therefore, the goal of our study is to further quantify the efficiency of monofilament mist nets for capturing bats when compared with regular mist nets. Due to the characteristics of monofilament mist nets, we expect that they will capture (i) a greater number of individuals and (ii) a greater diversity of species compared with regular mist nets. We also expect to capture (iii) a greater number of insectivorous bats in monofilament nets, given that they are known to effectively detect regular mist nets [16].

# 2. Material and methods

The study was conducted from 21 to 28 March, 2021 at La Cherenga Field Station located at Km 23, Guaycará district, Golfito, Costa Rica (8.639719° N, 83.074489° W). The station is located at 50 m above

(*a*)

(*b*)

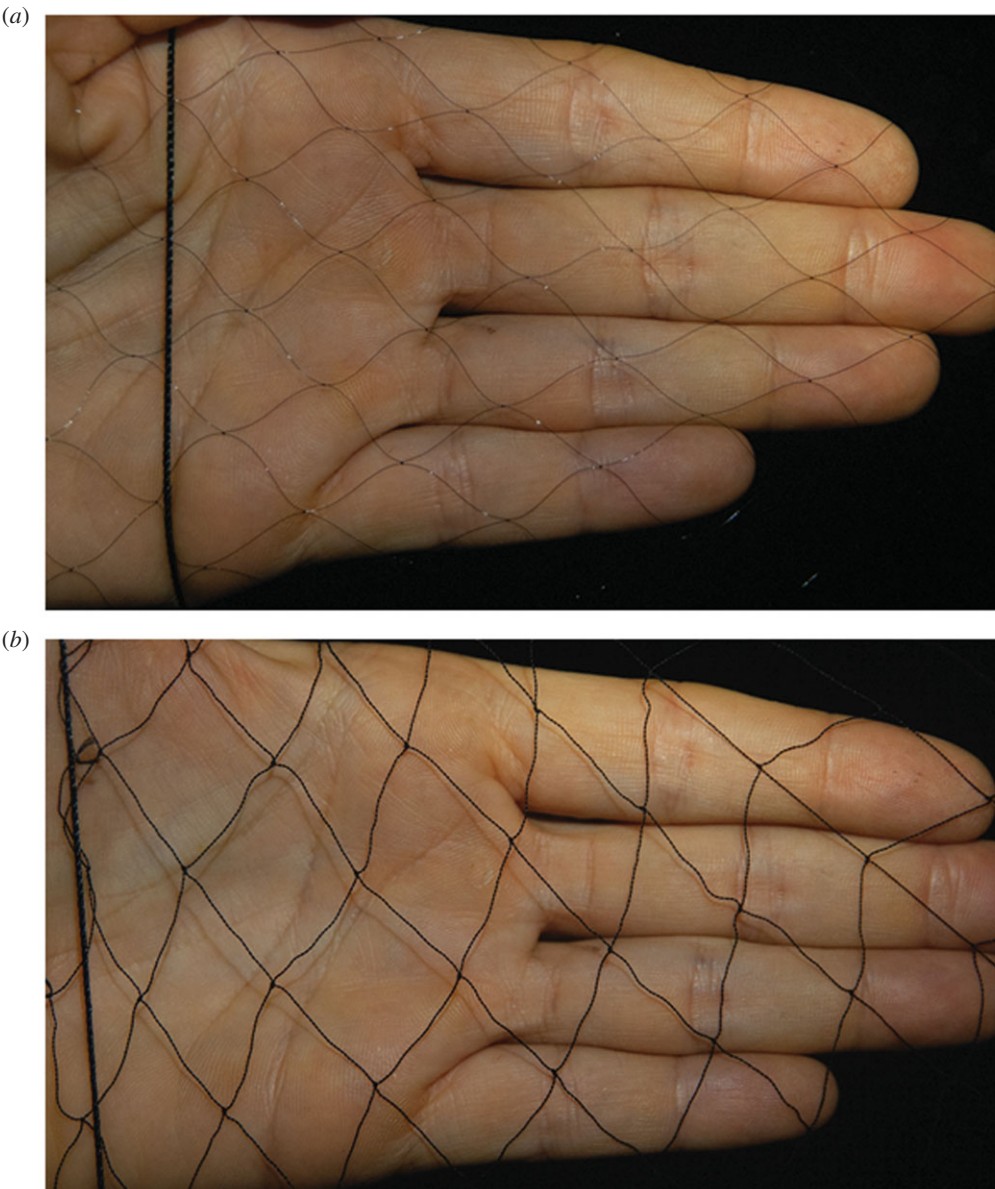

**Figure 1.** A monofilament net (*a*) compared with a regular net (*b*).

sea level, and its habitat corresponds to a tropical broad-leaved evergreen lowland forest with average temperatures ranging from 24 to 27°C and annual rainfall of approximately 5000 mm [20]. A large portion of the property is covered by forest in various stages of succession, in addition to an oil palm (*Elais guineensis*) plantation and grasslands. Our sampling was conducted for eight nights total and was alternated between habitat types, such that in total we sampled four nights in areas surrounded by forest (late secondary or primary) and four nights in the oil palm plantation.

We captured bats in mist nets (Ecotone, Poland) arranged in a block design; this block included a monofilament net and a regular nylon net (figure 1) placed next to each other in two possible configurations, depending on space availability, either in a straight line or in an L shape. The position of each type of net within the block was randomly selected. The nets ranged in length from 9 to 12 m; we always tried to place similarly sized nets in the same block, but an exact match was not always possible given the net sizes available to us. The monofilament nets we used were made with 0.08 mm single strand nylon, 14 mm mesh size, four shelves and were 2.4 m high. The regular nets were also made with nylon but of multiple strands, and they had a 19 mm mesh size, five shelves and were 3 m high. Most nets were placed at ground level, but during the last four nights, we placed a single block at approximately 5 m above the ground.

We opened nets between sunset at 18.00 until 22.00. Each block of two nets was monitored by a single person and was checked every 2 min to determine the presence of bats in them. If any individual was captured, it was promptly removed from the net to identify the species [21]. For each individual, the type of net in which it was captured, the species, sex, age and reproductive status were noted. Once all data were collected, the animal was released. All sampling protocols followed guidelines approved by the American Society of Mammalogists for capture, handling and care of mammals [22].

With the data collected, we estimated the number of bats captured per net per night. We implemented a correction of the capture effort by estimating the number of individuals captured per square metre per hour, as regular and monofilament nets did not have exactly the same size. We estimated species diversity based on Hill numbers in the package iNext [23], where $q = 0$ represents species richness, $q = 1$ represents the Shannon diversity index and $q = 2$ represents the Simpson diversity index. Increasing Hill numbers indicate a decreasing emphasis in the contribution of rare species to estimates of diversity [24]. Finally, we established the diet of each species sampled based on information gathered from previous studies [25].

We used R [26] and lme4 [27] to perform a linear mixed effects analysis of the relationship between the number of bats captured and type of net. As fixed effects, we entered type of net and habitat (without interaction term) into the model. As random effects, we included block and night. Visual inspection of residual plots did not reveal any obvious deviations from homoscedasticity or normality. $p$-values were obtained by likelihood ratio tests of the full model with the effect of net type and habitat against the model without the effect of net type, only including the effect of habitat. We also performed a chi-squared test to determine if there was a significant difference in the number of regular or monofilament nets that were able to capture insectivorous and frugivorous bats. Finally, we compared species accumulation curves (interpolated and extrapolated to double the number of captures for the net that captured the most bats) and sample coverage based on Hill numbers for both types of nets used. All code and raw data have been stored in the GitHub repository (https://github.com/morceglo/Monofilament-nets-for-bats).

## 3. Results

We placed a total of 46 nets throughout our eight-night sampling period, 23 regular and 23 monofilament, for a total of 738 and 585.6 m$^2$ sampled by regular and monofilament nets, respectively. Both types of nets were opened for 99.5 h, and in this period, we captured a total of 215 bats from 24 species (table 1); 90 individuals were captured in regular nets while 125 individuals were sampled using monofilament nets. Fourteen species were captured in regular nets and 20 species in monofilament nets. Also, 10 species were only captured in monofilament nets, such as several nectar-feeding bats (i.e. *Hylonycteris underwoodi* and *Lonchophylla concava*), several insectivorous species including many in the genus *Micronycteris* and others such as *Saccopteryx leptura* and *Thyroptera tricolor*. Only four species, *Chiroderma villosum*, *Desmodus rotundus*, *Myotis riparius* and *Trinycteris nicefori*, were solely captured in regular nets. The majority (11 out of 16) of rare species sampled, i.e. those that were captured in four or fewer occasions, are considered insectivorous (table 1).

We found that using monofilament nets affected the number of individuals captured ($\chi^2(1) = 6.47$, $p = 0.01$), increasing capture rates by about 0.04 bats per m$^2$ per hour compared with regular nets (figure 2). A significantly larger portion of insectivorous species (9 out of 11; table 1) was captured in monofilament nets compared with those captured in regular nets (4 out of 11; $\chi^2(1) = 4.70$, $p = 0.03$). Both types of nets were equally efficient at capturing frugivorous species (8 out of 9 species). Monofilament nets also appeared more efficient at capturing rare species than regular nets; 12 out of 16 rare species were captured in monofilament nets, whereas only 6 out 16 rare species were sampled using regular nets ($\chi^2(1) = 4.57$, $p = 0.03$).

In regular nets, species richness ($q = 0$) was estimated at 11 (s.e. = 4.45), while species richness estimated from captures made in monofilament nets was 20 (s.e. = 4.76). Similar differences were observed using other diversity indexes, including Shannon and Simpson ($q = 1$ and 2, respectively), although the difference in species diversity between net types decreased with increasing values in Hill numbers (figure 3). Species accumulation curves indicate that samples based on regular nets are significantly underestimating species diversity, even when sample sizes (i.e. individuals captured) are extrapolated to double the study's sampling effort. Also, the calculated sample coverage for both nets is equally large, suggesting that an inflated estimate of sample coverage is created by regular nets even though they seem to be underestimating species diversity (figure 3).

**Table 1.** Number of individuals captured per species in regular and monofilament nets.

| species | individuals captured | | feeding guild |
| --- | --- | --- | --- |
| | regular | monofilament | |
| *Artibeus jamaicensis* | 10 | 26 | frugivorous |
| *Artibeus lituratus* | 3 | 3 | frugivorous |
| *Carollia castanea* | 8 | 13 | frugivorous |
| *Carollia perspicillata* | 27 | 23 | frugivorous |
| *Carollia sowelli* | 9 | 6 | frugivorous |
| *Chiroderma villosum* | 3 | 0 | frugivorous |
| *Dermanura watsoni* | 17 | 21 | frugivorous |
| *Desmodus rotundus* | 1 | 0 | sanguinivorous |
| *Glossophaga soricina* | 1 | 7 | nectarivorous |
| *Hylonycteris underwoodi* | 0 | 1 | nectarivorous |
| *Lonchophylla concava* | 0 | 1 | nectarivorous |
| *Lophostoma brasiliense* | 0 | 2 | insectivorous |
| *Micronycteris hirsuta* | 0 | 2 | insectivorous |
| *Micronycteris microtis* | 1 | 2 | insectivorous |
| *Micronycteris minuta* | 0 | 1 | insectivorous |
| *Micronycteris schmidtorum* | 0 | 1 | insectivorous |
| *Myotis riparius* | 3 | 0 | insectivorous |
| *Peropteryx kappleri* | 0 | 1 | insectivorous |
| *Platyrrhinus helleri* | 0 | 4 | frugivorous |
| *Saccopteryx bilineata* | 1 | 3 | insectivorous |
| *Saccopteryx leptura* | 0 | 2 | insectivorous |
| *Thyroptera tricolor* | 0 | 1 | insectivorous |
| *Trinycteris nicefori* | 1 | 0 | insectivorous |
| *Uroderma bilobatum* | 5 | 5 | frugivorous |
| total | 90 | 125 | |

# 4. Discussion

In this study, we compared bat capture efficiency using regular and monofilament mist nets for sampling Neotropical bat communities. Our results demonstrate that monofilament nets were able to capture a larger number of individuals and species compared with regular nets. We found that bats of all trophic guilds were sampled by monofilament nets, yet these nets were particularly more effective at sampling insectivorous species compared with regular nets. We also show that monofilament nets were more likely to capture rare or elusive species and that this difference in capture rates was not affected by habitat type. Thus, net design, particularly with regard to the thickness of the strands used for constructing the netting, is an important factor to consider when the main interest of the study is to gauge species diversity or when certain trophic guilds, especially insectivorous, are targeted in bat surveys.

Conspicuousness of mist nets is considered to be a major factor contributing to lower capture rates in birds; thus, placing nets in the shade or against a dark background significantly increases their success [28,29]. While no studies to date have assessed how net placement affects capture rates in bats, our results of an overall increase in the number of sampled individuals strongly suggest that using monofilament nets may be an alternative and effective way to decrease their detection by bats. This decrease in the detectability of monofilament nets may also be largely responsible for an increased probability of capturing insectivorous species, which are known to be skilled at detecting and avoiding regular mist

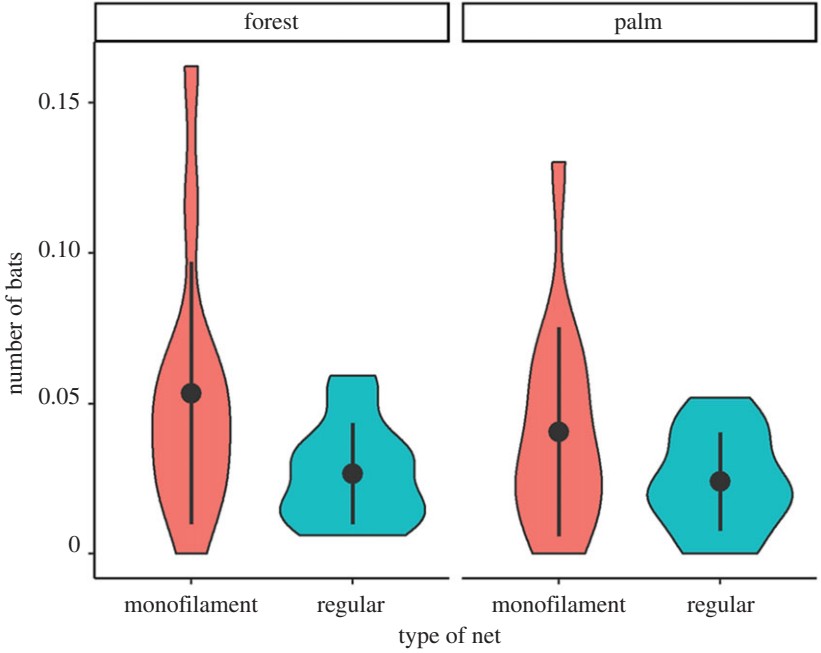

**Figure 2.** Violin plots showing the number of bats captured (corrected by sampling effort = bats m$^{-2}$ h$^{-1}$) according to the type of net used and the habitat sampled.

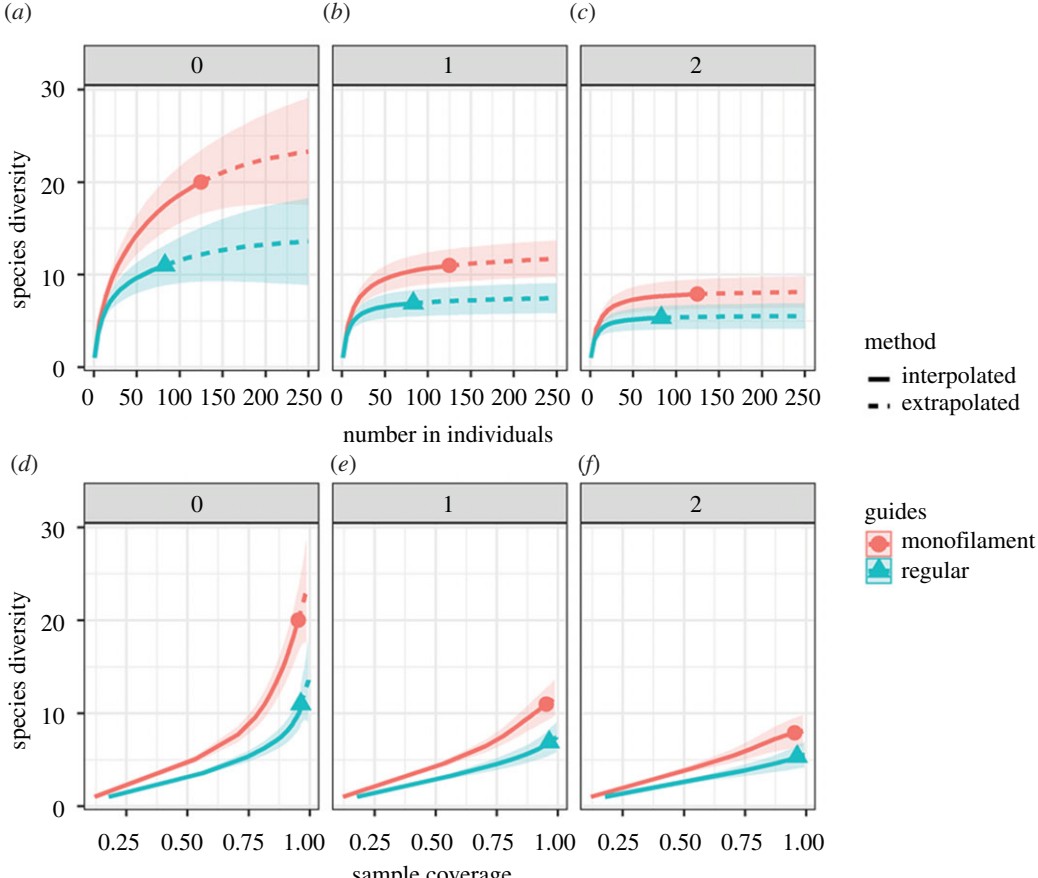

**Figure 3.** Species accumulation curves (a,b,c) and estimated sample coverage (d,e,f) for the two types of nets used. The numbers above panels represent the Hill numbers, where $q = 0$ represents species richness, $q = 1$ represents the Shannon diversity index and $q = 2$ represents the Simpson diversity index.

nets [16]. Since insectivorous bats comprised the majority of rare species sampled in our study, then it is expected that monofilament nets also captured a larger portion of rare species. With our data, however, we cannot confirm that these seldom-captured species are in fact locally rare, or if fewer individuals were captured because they remain skilled at avoiding even the monofilament nets. Despite the latter, monofilament nets were effective at detecting many rare species, particularly those in the genus *Micronycteris*, which may be missed in bat surveys that use acoustic detectors in addition to regular nets, as the former may not be able to record low-intensity echolocation calls [30,31].

The results of our study do not mirror those of Ferreira *et al.* [18], since they obtained the largest capture rates using regular, not monofilament, mist nets. Previous studies, however, indicate greater capture rates in monofilament nets. For example, a study by Chaverri *et al.* [32] shows that capture rates in regular, polyester, nets were 0.04 bats per net-hour, while capture rates in monofilament nets were 0.13 bats per net-hour. One potential problem with monofilament nets is that the strands break very easily, so they require constant revision as they can be easily and quickly damaged by bats. This may explain lower capture rates in monofilament nets in Ferreira's *et al.* study, as they checked nets approximately every 20 min, potentially allowing many bats to break the strands and escape the net before being tallied. Therefore, to more thoroughly seize the benefits of monofilament nets, we recommend checking time intervals between 3 and 5 min, which represents a shorter time interval compared with what is recommended for regular mist nets (15 min; [33,34]). Despite this, we consider that reducing net-checking intervals in general for bat extraction is a practice that should bring significant benefits, particularly regarding animal safety in addition to net capturing efficiency, as an increase in visits could reduce data loss due to escape or predation of bats in the net [33]. We are aware that applying these recommendations may require more field personnel, so researchers should carefully consider the benefits of using monofilament nets, in the form of a significant increase in capturing efficiency, and the costs involved in having additional personnel and an inevitable faster deterioration of their mist nets.

In conclusion, we show that incorporating monofilament nets into bat sampling designs offers not only an opportunity to expand records of different guilds and rare bat species, but ultimately may help to improve our understanding of poorly known species and assemblages while still using a relatively cheap and portable method. The use of monofilament nets could help compensate for the known limitations of regular nets (e.g. inefficient at trapping insectivorous bats) and even harp traps (e.g. reduced portability and sampling area), providing an additional tool for the study of bat species. Additional studies are needed to understand the functionality of monofilament nets for other bat assemblages; meanwhile, we advocate for the use of this simple tool and ideally in combination with others, including regular mist nets, but also acoustic and roost surveys.

Ethics. All protocols for capturing and handling bats comply with the current laws of the Costa Rican government (permit no. ACOSA-D-R-056-2021).

Data accessibility. Data and relevant code for this research work are stored in GitHub: [https://github.com/morceglo/Monofilament-nets-for-bats] and have been archived within the Zenodo repository: [https://zenodo.org/badge/latestdoi/364426188].

Author's contributions. All authors participated in the design of the study, collected field data and drafted the manuscript. G.C. conducted statistical analyses. All authors gave final approval for publication.

Competing interests. We have no competing interests.

Funding. We received no funding for this study.

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
