## [Peer Review File · Royal Society Open Science]

Review History

RSOS-211404.R0 (Original submission)

Review form: Reviewer 1 (Angelo Soto-Centeno)

Is the manuscript scientifically sound in its present form?

Yes

Are the interpretations and conclusions justified by the results?

Yes

Is the language acceptable?

Yes

Do you have any ethical concerns with this paper?

No

Have you any concerns about statistical analyses in this paper?

No

Recommendation?

Accept with minor revision (please list in comments)

Comments to the Author(s)

This manuscript focuses on testing a classical and popular method for capturing bats in the field: mist nets. The authors compared regular and monofilament mist nets to determine if there is a difference in capture efficiency between each. The capture efficiency was quantified as differences in species richness and species diversity. The authors found that monofilament mist nets can sample greater species richness and diversity, particularly of rare bat species, compared to regular mist nets. The study was well designed, all statistical comparisons were appropriate, and the figures help convey the punchline of the paper. I have used mist nets for capturing bats myself for over 20 years and despite that in recent years I have seen many colleagues bring a few monofilament mist nets to the field claiming that they can sample bats better, I never saw a direct comparison testing this hypothesis until now. I believe this study has great applicability for those seeking to improve bat bioinventories and regional natural history accounts, but also can help field biologists make decisions on the type of field gear needed. Beyond this, given that thorough bioinventories are often needed to generate species management plans for many bats, I think this study can also be important for those studying bat conservation.

I commend the authors for submitting a very well written piece that is clear and nearly void of grammatical errors. There are only a few instances where clarification of the text is suggested (see the list of line items below). The introduction of the problem examined is well laid out and logical. And the discussion properly addresses the results and explores their significance even providing a good representation of caveats for the capture method they are espousing. The study is simple and straight forward, and I do not see any technical issues that need to be raised. Please see my list of suggestions below.

Specific line points that could use a quick revision.

Abstract

Line 23: "Here we compare capture efficiency ..." - consider adding a comma after "Here, ..."

Introduction

Line 47: "... interpretation, which thus deteriorates ..." - delete the word "thus".

Line 59: "A wide variety of methods exists ..." - delete the "s" in the word "exists".

Line 61: "... methods for capturing and handling bats ..." - delete "and handling".

Line 62: "On the other hand, indirect ..." - delete "On the other hand, ".

Line 65: "... are considered more invasive ..." - delete "more ", it is not necessary as you did not include other methods in the sentence as comparison.

Line 66: "... previous experience, especially during ..." - perhaps insert "by the user" to improve clarity.

Line 67: "... ; however, they are essential ..." - use a period before "however" instead of semicolon to shorten the long running sentence.

Line 71: "... detect and therefore avoid ..." - delete "therefore".

Line 71-72: "... other trapping methods have been developed ..." - Some might argue that harp-traps were developed for capturing bats under different habitat circumstances instead of as a replacement for mist-nets. Truly, the sampling area of a harp-trap is much smaller than that of most mist-nets. So, these two cannot be directly compared. Maybe this fragment can be deleted and the sentence reworded a bit to avoid that direct comparison.

Results

Line 187: Figure 3 caption - I may have missed this... It seems like panels 0, 1, and 2 correspond to species richness, Shannon and Simpson diversity indices. It would make it more clear if this was added to the figure caption.

Discussion

Line 214: "... record low intensity calls." - insert the word "echolocation" after intensity to improve clarity.

Cheers,

Dr. Angelo Soto-Centeno

Review form: Reviewer 2

Is the manuscript scientifically sound in its present form?

Yes

Are the interpretations and conclusions justified by the results?

Yes

Is the language acceptable?

Yes

Do you have any ethical concerns with this paper?

No

Have you any concerns about statistical analyses in this paper?

No

Recommendation?

Accept with minor revision (please list in comments)

Comments to the Author(s)

Dear authors,

you present a field study comparing 2 types of mistnets that are used for catching bats. I only have 2 major points and some minor (see Appendix A):

1. Please clarify what you mean by traditional/regular mist nets. There are many types of mist nets that are used "traditionally" or "regularly" depending on the region of the world. Finding appropriate terms would be helpful.

2. Please discuss a recently published work by Fereirra et al. 2021, Are bat mist nets ideal for capturing bats? From ultrathin to bird nets, a field test. Journal of Mammalogy. DOI: 10.1093/jmammal/gyab109.

Decision letter (RSOS-211404.R0)

Dear Miss Chaves-Ramírez

On behalf of the Editors, we are pleased to inform you that your Manuscript RSOS-211404 "Comparing the efficiency of monofilament and traditional nets for capturing bats" has been accepted for publication in Royal Society Open Science subject to minor revision in accordance with the referees' reports. Please find the referees' comments along with any feedback from the Editors below my signature.

Please submit your revised manuscript and required files (see below) no later than 7 days from today's (ie 22-Oct-2021) date. Note: the ScholarOne system will 'lock' if submission of the revision is attempted 7 or more days after the deadline. If you do not think you will be able to meet this deadline please contact the editorial office immediately.

on behalf of Professor Marcelo Sanchez (Associate Editor) and Pete Smith (Subject Editor)
openscience@royalsociety.org

Associate Editor Comments to Author (Professor Marcelo Sanchez):

Comments to the Author:

Considering the specific and constructive comments by the reviewers should improve the manuscript

Reviewer comments to Author:

Reviewer: 1

Comments to the Author(s)

This manuscript focuses on testing a classical and popular method for capturing bats in the field: mist nets. The authors compared regular and monofilament mist nets to determine if there is a difference in capture efficiency between each. The capture efficiency was quantified as differences in species richness and species diversity. The authors found that monofilament mist nets can sample greater species richness and diversity, particularly of rare bat species, compared to regular mist nets. The study was well designed, all statistical comparisons were appropriate, and the figures help convey the punchline of the paper. I have used mist nets for capturing bats myself for over 20 years and despite that in recent years I have seen many colleagues bring a few monofilament mist nets to the field claiming that they can sample bats better, I never saw a direct comparison testing this hypothesis until now. I believe this study has great applicability for those seeking to improve bat bioinventories and regional natural history accounts, but also can help

field biologists make decisions on the type of field gear needed. Beyond this, given that thorough bioinventories are often needed to generate species management plans for many bats, I think this study can also be important for those studying bat conservation.

I commend the authors for submitting a very well written piece that is clear and nearly void of grammatical errors. There are only a few instances where clarification of the text is suggested (see the list of line items below). The introduction of the problem examined is well laid out and logical. And the discussion properly addresses the results and explores their significance even providing a good representation of caveats for the capture method they are espousing. The study is simple and straight forward, and I do not see any technical issues that need to be raised. Please see my list of suggestions below.

Specific line points that could use a quick revision.

Abstract

Line 23: "Here we compare capture efficiency ..." - consider adding a comma after "Here, ..."

Introduction

Line 47: "... interpretation, which thus deteriorates ..." - delete the word "thus".

Line 59: "A wide variety of methods exists ..." - delete the "s" in the word "exists".

Line 61: "... methods for capturing and handling bats ..." - delete "and handling".

Line 62: "On the other hand, indirect ..." - delete "On the other hand, ".

Line 65: "... are considered more invasive ..." - delete "more ", it is not necessary as you did not include other methods in the sentence as comparison.

Line 66: "... previous experience, especially during ..." - perhaps insert "by the user" to improve clarity.

Line 67: "... ; however, they are essential ..." - use a period before "however" instead of semicolon to shorten the long running sentence.

Line 71: "... detect and therefore avoid ..." - delete "therefore".

Line 71-72: "... other trapping methods have been developed ..." - Some might argue that harp-traps were developed for capturing bats under different habitat circumstances instead of as a replacement for mist-nets. Truly, the sampling area of a harp-trap is much smaller than that of most mist-nets. So, these two cannot be directly compared. Maybe this fragment can be deleted and the sentence reworded a bit to avoid that direct comparison.

Results

Line 187: Figure 3 caption - I may have missed this... It seems like panels 0, 1, and 2 correspond to species richness, Shannon and Simpson diversity indices. It would make it more clear if this was added to the figure caption.

Discussion

Line 214: "... record low intensity calls." - insert the word "echolocation" after intensity to improve clarity.

Cheers,

Dr. Angelo Soto-Centeno

Reviewer: 2

Comments to the Author(s)

Dear authors,

you present a field study comparing 2 types of mistnets that are used for catching bats. I only have 2 major points and some minor (see PDF):

1. Please clarify what you mean by traditional/regular mist nets. There are many types of mist nets that are used "traditionally" or "regularly" depending on the region of the world. Finding appropriate terms would be helpful.
2. Please discuss a recently published work by Fereirra et al. 2021, Are bat mist nets ideal for capturing bats? From ultrathin to bird nets, a field test. *Journal of Mammalogy*. DOI: 10.1093/jmammal/gyab109.

===PREPARING YOUR MANUSCRIPT===

one version should clearly identify all the changes that have been made (for instance, in coloured highlight, in bold text, or tracked changes);

===PREPARING YOUR REVISION IN SCHOLARONE===

Please ensure that you include a summary of your paper at the 'Type, Title, & Abstract' step. This should be no more than 100 words to explain to a non-scientific audience the key findings of your

research. This will be included in a weekly highlights email circulated by the Royal Society press office to national UK, international, and scientific news outlets to promote your work. An effective summary can substantially increase the readership of your paper.

-- If you are requesting an article processing charge waiver, you must select the relevant waiver option (if requesting a discretionary waiver, the form should have been uploaded, see 'File upload' above).

-- If you have uploaded any electronic supplementary (ESM) files, please ensure you follow the guidance at <https://royalsociety.org/journals/authors/author-guidelines/#supplementary-material> to include a suitable title and informative caption. An example of appropriate titling and captioning may be found at https://figshare.com/articles/Table_S2_from_Is_there_a_trade-off_between_peak_performance_and_performance_breadth_across_temperatures_for_aerobic_scope_in_teleost_fishes_/3843624.

Author's Response to Decision Letter for (RSOS-211404.R0)

See Appendix B.

Decision letter (RSOS-211404.R1)

Dear Dr Chaverri,

I am pleased to inform you that your manuscript entitled "Comparing the efficiency of monofilament and regular nets for capturing bats" is now accepted for publication in Royal Society Open Science.

on behalf of Professor Marcelo Sanchez (Associate Editor) and Pete Smith (Subject Editor)
openscience@royalsociety.org

Appendix A**ROYAL SOCIETY
OPEN SCIENCE****Comparing the efficiency of monofilament and traditional
nets for capturing bats**

Journal:	Royal Society Open Science
Manuscript ID	RSOS-211404
Article Type:	Research
Date Submitted by the Author:	29-Aug-2021
Complete List of Authors:	Chaves-Ramírez, Silvia; Costa Rica, Escuela de Biología Castillo-Salazar, Christian; Costa Rica, Escuela de Biología Sánchez-Chavarría, Mariela; Costa Rica, Sede del Sur Solís-Hernández, Hellen; Costa Rica, Sede del Sur Chaverri, Gloriana; Universidad de Costa Rica, Sede del Sur; Smithsonian Tropical Research Institute,
Subject:	ecology < BIOLOGY
Keywords:	bats, Chiroptera, Costa Rica, mist nets
Subject Category:	Ecology, Conservation, and Global Change Biology

Author-supplied statements

Relevant information will appear here if provided.

Ethics

Does your article include research that required ethical approval or permits?:

Yes

Statement (if applicable):

All protocols for capturing and handling bats comply with the current laws of the Costa Rican government (permit no. ACOSA-D-R-056-2021).

Data

It is a condition of publication that data, code and materials supporting your paper are made publicly available. Does your paper present new data?:

Yes

Statement (if applicable):

All code and raw data have been stored in the GitHub repository (<https://github.com/morceglo/Monofilament-nets-for-bats>).

Conflict of interest

I/We declare we have no competing interests

Statement (if applicable):

CUST_STATE_CONFLICT :No data available.

Authors' contributions

This paper has multiple authors and our individual contributions were as below

Statement (if applicable):

All authors participated in the design of the study, collected field data, drafted the manuscript. G.C. conducted statistical analyses. All authors gave final approval for publication.

Title: **Comparing the efficiency of monofilament and traditional nets for capturing bats**

Authors: Silvia Chaves-Ramírez¹, Christian Castillo-Salazar¹, Mariela Sánchez-Chavarría², Hellen
Solís-Hernández², Gloriana Chaverri^{2,3*}

ORCID iDs: Christian Castillo-Salazar 0000-0003-0468-983X, Gloriana Chaverri 0000-0002-1155-
432X

Affiliations: 1 Escuela de Biología, Universidad de Costa Rica, San Pedro, Costa Rica; 2 Sede del
Sur, Universidad de Costa Rica, Golfito, Costa Rica; 3 Smithsonian Tropical Research Institute,
Balboa, Ancón, Panamá

*Author for correspondence. E-mail: gloriana.chaverri@ucr.ac.cr

Abstract

Traditional mist nets used for capturing bats have several drawbacks, particularly that they are inefficient at sampling many insectivorous species. One possible alternative is to use monofilament nets, whose netting is made of single strands of yarn instead of several as regular nets, making them less detectable. To date, no study has quantified the capture efficiency of monofilament nets compared to regular mist nets in the study of bats. Here we compare capture efficiency of monofilament and regular mist nets, focusing on bat abundance and species diversity at a lowland tropical forest in southwestern Costa Rica. During our sampling period, we captured 90 individuals and 14 species in regular nets and 125 individuals and 20 species in monofilament nets. The use of monofilament nets increased overall capture rates, but most notably for insectivorous species. Species accumulation curves indicate that samples based on regular nets are significantly underestimating species diversity, most notably as these nets fail at sampling rare species. We show that incorporating monofilament nets into bat studies offers an opportunity to expand records of different guilds and rare bat species and to improve our understanding of poorly-known bat assemblages while using a popular, relatively cheap and portable sampling method.

Key words: bats, Chiroptera, Costa Rica, mist nets.

**Introduction**

In animal research studies it is often necessary to capture the study organism, and the sample of
individuals that are trapped should ideally be representative of the target species or assemblage.
However, there are many problems associated with trapping methods that hamper the quality of data
obtained, most notably differences in capturing success among individuals or species. A well-known
phenomenon, for example, is the difference in trap-shy or trap-happy species, or those where
individuals consistently avoid traps or those which consistently seek them, respectively, which
creates serious sampling biases [1]. Also, some individuals within a species are easily trapped while
others are not [2], and some sampling methods are consistently more effective than others, even
within the same taxon [3,4]. These biases may significantly affect a study's results and their
interpretation, which thus deteriorates the decisions involving the taxon under study or its habitats.
For example, capture methods may bias estimates of density and population structure [2], population
trends [5], and estimates of species richness and capture rates [6]. It is still possible to account for
capture probabilities as a way to correct for some of these biases [2], yet this requires thorough
knowledge of a trapping method's effect on all target individuals and species [5,7], data which are
commonly not available for the majority of species. Therefore, selecting an appropriate and
representative trapping method is critical for making the correct inferences and appropriate decisions
concerning the focal organism.

Bats are the most widely distributed terrestrial mammals on Earth and constitute almost one-
fifth of mammalian biodiversity [8]. However, their ability to fly and nocturnal habits make them a
difficult group to study [9]; thus, efficient sampling methods are essential for their capture and
identification. Despite their large diversity and wide distribution, information on many species is still
deficient, especially in areas that harbor the greatest diversity [10]. A wide variety of methods exists
for the study of bats, which differ in effectiveness and practicality depending on the goal of the

61 study. Among the most commonly known methods for capturing and handling bats are mist nets,
harp traps, hand nets, and direct captures at their roosts [11]. On the other hand, indirect recording of
species has increased with the use of camera traps, thermal cameras and acoustic recording
equipment [12].

Bat capture methods, such as mist nets, are considered **more invasive** they increase the
stress associated with the capture process and require substantial previous experience, especially
during the process of extracting individuals from the net [13,14]; however, they are essential for
collecting information on morphometrics, **acoustics** sensitivity to ectoparasites, species diversity,
and **population health**, among others. An additional problem with the use of mist nets is that some
species are very difficult to capture, either because they fly extremely high or because their sensory
abilities allow them to detect and therefore avoid the nets [15,16]. Therefore, other trapping methods
have been developed, such as harp traps, which tend to reflect fewer high-frequency echoes
compared to traditional mist nets, making it easier to trap species that use high-frequency calls,
especially those that feed on insects [17–19]. But harp traps also have limitations; for example, their
sampling area is small, making it necessary to increase the number of traps and to have prior
knowledge of flight routes to increase capture efficiency. On the other hand, indirect methods such
as acoustic monitoring may be advantageous since they do not cause stress to the animals and may
be able to record individuals that fly very high and/or are difficult to capture. Acoustic monitoring
studies, however, depend on the availability of sufficiently large and representative acoustic libraries,
which are not yet available for many sites and species [20]. In addition, the use of acoustic recording
equipment imposes higher economic costs than other existing capture methods

Despite the difficulties described above, mist nets remain an essential, practical and
accessible method for capturing bats, facilitating research and monitoring of species worldwide.
Monofilament nets were developed as a novel tool for bat trapping, and these could have great

potential to minimize some of the detectability limitations of using mist nets, while still benefiting
from the already well-described advantages of this trapping method. Monofilament nets are designed
to be less detectable by bats, as their netting is made of single strands of yarn, unlike traditional nets
whose netting is created by twisting several individual strands. However, to date no study has
quantified their capture efficiency compared to traditional mist nets. Therefore, the goal of our study
is to quantify the efficiency of monofilament mist nets for capturing bats when compared to
traditional nylon mist nets. Due to the characteristics of monofilament mist nets, we expect that they
will capture i) a greater number of individuals, and ii) a greater diversity of species compared to
traditional mist nets. We also expect to capture iii) a greater number of insectivorous bats in
monofilament nets, given that they are known to effectively detect traditional mist nets.
**Materials and methods**

[revised manuscript text omitted]

Monofilament nets also appeared more efficient at capturing rare species than regular nets; 12 out of
 16 rare species were captured in monofilament nets, whereas only 6 out of 16 rare species were
 sampled using regular nets ($\chi^2(1) = 4.57, p = 0.03$).

Figure 2. Violin plots showing the number of bats captured (corrected by sampling effort = bats per
 174 m² per hour) according to the type of net used and the habitat sampled.

In regular nets, species richness ($q = 0$) was estimated at 11 (s.e. = 4.45), while species
 richness estimated from captures made in monofilament nets was 20 (s.e. = 4.76). Similar differences
 were observed using other diversity indexes, including Shannon and Simpson ($q = 1$ & 2,
 respectively), although the difference in species diversity between net types decreased with
 increasing values in Hill numbers (Figure 3). Species accumulation curves indicate that samples
 based on regular nets are significantly underestimating species diversity, even when sample sizes
 (i.e., individuals captured) are extrapolated to double the study's sampling effort. Also, the calculated

sample coverage for both nets is equally large, suggesting that an inflated estimate of sample coverage is created by regular nets even though they seem to be underestimating species diversity (Figure 3).

Figure 3. Species accumulation curves (upper panels) and estimated sample coverage (lower panels) for the two types of nets used.

Discussion

In this study we compared bat capture efficiency using regular and monofilament mist nets for sampling Neotropical bat communities. Our results demonstrate that monofilament nets were able to capture a larger number of individuals and species compared to regular nets. We found that bats of all trophic guilds were sampled by monofilament nets, yet these nets were particularly more effective at

195 sampling insectivorous species compared to regular nets. We also show that monofilament nets were
196 more likely to capture rare or elusive species and that this difference in capture rates was not affected
by habitat type. Thus, net design, particularly with regards to the **number of strands used for**
**198 constructing the netting,**  an important factor to consider when the main interest of the study is to
gauge species diversity or when certain trophic guilds, especially insectivorous, are targeted in bat
surveys.

Conspicuousness of mist nets is considered to be a major factor contributing to lower capture
rates in birds; thus, placing nets in the shade or against a dark background significantly increases their
success [29,30]. While no studies to date have assessed how net placement affects capture rates in
bats, our results of an overall increase in the number of sampled individuals strongly suggest that using
monofilament nets may be an alternative and effective way to decrease their detection by bats. This
decrease in detectability of monofilament nets may also be largely responsible for an increased
probability of capturing insectivorous species, which are known to be skilled at detecting and avoiding
regular mist nets [16]. Since insectivorous bats comprised the majority of rare species sampled in our
study, then it is expected that monofilament nets also captured a larger portion of rare species. With
our data, however, we cannot confirm that these seldom-captured species are in fact locally rare, or if
fewer individuals were captured because they remain skilled at avoiding even the monofilament nets.
Despite the latter, monofilament nets were effective at detecting many rare species, particularly those
in the genus *Micronycteris*, which may be missed in bat surveys that use acoustic detectors in addition
to regular nets, as the former may not be able to record low-intensity calls [31,32].

While we found no studies to date that have explicitly compared the efficiency in capture rates
between regular and monofilament nets using a paired design like ours, some previous results indicate
greater capture rates in the latter. For example, a study by Chaverri et al. (2016) shows that capture
rates in regular, polyester, nets were 0.04 bats per net-hour, while capture rates in monofilament nets
were 0.13 bats per net-hour. Other studies in birds suggest that both mesh size, color and the number

of strands that form the netting influence capture rates of nets placed over fish ponds, with
monofilament nets imposing greater risks [34]. The previous results are to be expected, as detectability
of capturing devices clearly affects their effectiveness. However, our results further demonstrate that
the netting material may affect the species that are sampled through mist-nets.

It is important to emphasize that before deciding to use monofilament nets in a study, all
researchers involved must be skillful at removing bats from mist nets, as the strands are often hard to
see and they break very easily, monofilament nets thus require constant revision, since they can be
easily and quickly damaged by bats. Therefore, we recommend checking time intervals between 3 to
5 minutes, which represents a shorter time interval compared to what's recommended for regular mist-
nets (15 min; (Gannon and Sikes 2007; Kunz et al. 2009). Despite this, we consider that reducing net-
checking intervals in general for bat extraction is a practice that should bring significant benefits,
particularly regarding animal safety and net capturing efficiency, as an increase in visits could reduce
data loss due to escape or predation of bats in the net [35]. We are aware that applying these
recommendations may require more field personnel, so researchers should carefully consider the
benefits of using monofilament nets, in the form of a significant increase in capturing efficiency, and
the costs involved in having additional personnel and an inevitable faster deterioration of their mist
nets.

In conclusion, we show that incorporating monofilament nets into bat sampling designs offers
not only an opportunity to expand records of different guilds and rare bat species, but ultimately may
help to improve our understanding of poorly-known species and assemblages while still using a
relatively cheap and portable method. The use of monofilament nets could help compensate for the
known limitations of regular nets (e.g., inefficient at trapping insectivorous bats) and even harp traps
(e.g., reduced portability and sampling area), providing an additional tool for the study of bat species.
Additional studies are needed to understand the functionality of monofilament nets for other bat

assemblages; meanwhile, we advocate for the use of this simple tool and ideally in combination with
others, including regular mist nets, but also acoustic and roost surveys.

Ethics statement. All protocols for capturing and handling bats comply with the current laws of the
Costa Rican government (permit no. ACOSA-D-R-056-2021).

Data accessibility. All code and raw data have been stored in the GitHub repository
(<https://github.com/morceglo/Monofilament-nets-for-bats>).

Author's contributions. All authors participated in the design of the study, collected field data,
drafted the manuscript. G.C. conducted statistical analyses. All authors gave final approval for
publication.

Competing interests. We have no competing interests.

References

1. Biro PA, Dingemanse NJ. 2009 Sampling bias resulting from animal personality. *Trends Ecol. Evol.* **24**, 66–67. (doi:10.1016/j.tree.2008.11.001)
2. Bisi F, Newey S, Nodari M, Wauters LA, Harrison A, Thirgood S, Martinoli A. 2011 The strong and the hungry: bias in capture methods for mountain hares *Lepus timidus*. *Wildlife Biol.* **17**, 311–316. (doi:10.2981/10-133)
3. Williams DF, Braun SE. 1983 Comparison of Pitfall and Conventional Traps for Sampling Small Mammal Populations. *J. Wildl. Manage.* **47**, 841. (doi:10.2307/3808622)

- 4. Flaquer C, Torre I, Arrizabalaga A. 2007 Comparison of sampling methods for inventory of
bat communities. *J. Mammal.* **88**, 526–533.
- 5. Garshelis DL, Noyce K V. 2006 Discerning Biases in a Large Scale Mark-Recapture
Population Estimate for Black Bears. *J. Wildl. Manage.* **70**, 1634–1643.
- 6. Kolowski JM, Forrester TD. 2017 Camera trap placement and the potential for bias due to
trails and other features. *PLoS One* **12**, e0186679. (doi:10.1371/journal.pone.0186679)
- 7. Baker PJ, Harris S, Robertson CPJ, Saunders G, White PCL. 2001 Differences in the capture
rate of cage-trapped red foxes *Vulpes vulpes* and an evaluation of rabies control measures in
Britain. *J. Appl. Ecol.* **38**, 823–835. (doi:10.1046/j.1365-2664.2001.00637.x)
- 8. Burgin CJ, Colella JP, Kahn PL, Upham NS. 2018 How many species of mammals are there?
*J. Mammal.* **99**, 1–14. (doi:10.1093/jmammal/gyx147)
- 9. Voigt CC, Kingston T. 2015 *Bats in the anthropocene: Conservation of bats in a changing*
*world*. (doi:10.1007/978-3-319-25220-9)
- 10. Frick WF, Kingston T, Flanders J. 2020 A review of the major threats and challenges to global
bat conservation. *Ann. N. Y. Acad. Sci.* **1469**, 5–25. (doi:10.1111/nyas.14045)
- 11. Wilson DE, Cole FR, Nichols JD, Rudran R, Foster MS. 1996 *Measuring and monitoring*
*biological diversity: Standard methods for mammals*. Washington, D.C.: Smithsonian Books.
See <http://pubs.er.usgs.gov/publication/5200145>.
- 12. Kunz TH, Parsons S. 2009 *Ecological and behavioral methods for the study of bats*.
Baltimore, MD: John Hopkins University Press.
- 13. Rizo-Aguilar A, Ávila-Torresagaton LG, Fuentes Vargas L, Lara Nuñez AC, Flores Nuñez
GI, Albino Miranda S. 2015 Técnicas para el estudio de los murciélagos. In *Manual de*
*técnicas del estudio de la fauna* (ed S Gallina Tessaro), Veracruz, México: Instituto de

Ecología, A. C.
14. Kunz TH, Kurta A. 1988 Capture methods and holding devices. In *Ecological and Behavioral*
*Methods for the Study of Bats* (ed TH Kunz), pp. 1–29. Washington: Smithsonian Institution
Press.
15. Kalko EK V., Estrada Villegas S, Schmidt M, Wegmann M, Meyer CFJ. 2007 Flying high--
assessing the use of the aerosphere by bats. *Integr. Comp. Biol.* **48**, 60–73.
(doi:10.1093/icb/icn030)
16. MacSwiney MC, Clarke FM, Racey PA. 2008 What you see is not what you get: The role of
ultrasonic detectors in increasing inventory completeness in Neotropical bat assemblages. *J.*
*Appl. Ecol.* **45**, 1364–1371. (doi:10.1111/j.1365-2664.2008.01531.x)
17. Pech-Canche JM, Estrella E, López-Castillo DL, Hernández-Betancourt SF, Moreno CE. 2011
Complementarity and efficiency of bat capture methods in a lowland tropical dry forest of
Yucatán, Mexico. *Rev. Mex. Biodivers.* **82**. (doi:10.22201/ib.20078706e.2011.3.683)
18. Berry N, O’connor W, Holderied MW, Jones G. 2004 Detection and Avoidance of Harp Traps
by Echolocating Bats. *Acta Chiropterologica* **6**, 335–346. (doi:10.3161/001.006.0211)
19. Duffy AM, Lumsden LF, Caddle CR, Chick RR, Newell GR. 2000 The efficacy of AnaBat
ultrasonic detectors and harp traps for surveying microchiropterans in south-eastern Australia.
*Acta Chiropterologica* **2**, 127–144.
20. Hayes JP. 2000 Assumptions and practical considerations in the design and interpretation of
echolocation-monitoring studies. *Acta Chiropterologica* **2**, 225–236.
21. Kappelle M, Castro M, Acevedo H, Gonzalez L, Monge H. 2002 *Ecosistemas del Area de*
*Conservacion Osa (ACOSA)*. Santo Domingo de Heredia, Costa Rica: Instituto Nacional de
Biodiversidad.

- 22. York HA, Rodríguez-Herrera B, LaVal RK, Timm RM, Lindsay KE. 2019 Field key to the
bats of Costa Rica and Nicaragua. *J. Mammal.* **100**, 1726–1749.
(doi:10.1093/jmammal/gyz150)
- 23. Sikes RS. 2016 2016 Guidelines of the American Society of Mammalogists for the use of wild
mammals in research and education. *J. Mammal.* **97**, 663–688.
(doi:10.1093/jmammal/gyw078)
- 24. Hsieh TC, Ma KH, Chao A. 2016 iNEXT: An R package for rarefaction and extrapolation of
species diversity (Hill numbers). *Methods Ecol. Evol.* **in press**, 1–22.
- 25. Chao A, Gotelli NJ, Hsieh TC, Sander EL, Ma KH, Colwell RK, Ellison AM. 2014
Rarefaction and extrapolation with Hill numbers: A framework for sampling and estimation in
species diversity studies. *Ecol. Monogr.* **84**, 45–67. (doi:10.1890/13-0133.1)
- 26. Kalko EKV, Handley CO, Handley D. 1996 Organization, diversity, and long-term dynamics
of a neotropical bat community. In *Long-term Studies in Vertebrate Communities* (eds M
Codyand, J Smallwood), pp. 503–553. Los Angeles, California: Academic Press.
- 27. R Core T. 2018 R: A language and Environment for Statistical Computing. *Vienna, Austria*
- 28. Bates D, Mächler M, Bolker BM, Walker SC. 2015 Fitting Linear Mixed-Effects Models
Using lme4. *J. Stat. Softw.* **67**, 1–48. (doi:10.18637/jss.v067.i01)
- 29. Jenni L, Leuenberger M, Rampazzi F. 1996 Capture Efficiency of Mist Nets with Comments
on Their Role in the Assessment of Passerine Habitat Use. *J. F. Ornithol.* **67**, 263–274.
- 30. Keyes BE, Grue CE. 1982 Capturing birds with mist nets: a review. *North Am. Bird Bander* **7**,
2–14.
- 31. O’Farrell MJ, Gannon WL. 1999 A comparison of acoustic versus capture techniques for the
inventory of bats. *J. Mammal.* **80**, 24–30.

- 32. Portfors C V. *et al.* 2000 Bats from Fazenda Intervales, Southeastern Brazil: species account
and comparison between different sampling methods. *Rev. Bras. Zool.* **17**, 533–538.
(doi:10.1590/S0101-81752000000200022)
- 33. Chaverri G, Garin I, Alberdi A, Jimenez L, Castillo-Salazar C, Aihartza J. 2016 Unveiling the
hidden bat diversity of a neotropical montane forest. *PLoS One* **11**.
(doi:10.1371/journal.pone.0162712)
- 34. Nemptzov SC, Olsvig-Whittaker L. 2003 The Use of Netting Over Fishponds as a Hazard to
Waterbirds. *Waterbirds* **26**, 416–423.
- 35. Kunz TH, Hodgkison R, Weise CD. 2009 Methods of capturing and handling bats. In
Ecological and Behavioral Methods for the Study of Bats (eds TH Kunz, S Parsons), pp. 3–35.
Baltimore, Maryland: The Johns Hopkins University Press.
- 36. Gannon WL, Sikes RS. 2007 Guidelines of the American Society of Mammalogists for the
Use of Wild Mammals in Research. *J. Mammal.* **88**, 809–823. (doi:10.1644/06-MAMM-F-
185R1.1)

Appendix B

October 28, 2021

Professor Jeremy Sanders
Editor-In-Chief
Royal Society Open Access

Dear Dr. Sanders,

We are resubmitting the manuscript titled “Comparing the efficiency of monofilament and traditional nets for capturing bats” with the changes suggested by the two reviewers. Specifically, we incorporated all the suggestions made by Dr. Angelo Soto-Centeno (Reviewer 1). As per the suggestions made by Reviewer 2, we also attempted to clarify in the abstract, introduction and methods what we mean by “traditional mist nets”, and have cited the recently-published paper by Ferreira et al. in the introduction and discussion. We have also addressed all comments made by Reviewer 2 in the pdf.

Below, we detail how we have addressed each comment/suggestion made by each reviewer. We address each separate comment/suggestion with our response in bold. Whenever relevant, we also include a section in italics noting how we have redacted the sentence(s) as per the reviewers’ suggestion.

We hope you will find the changes to our manuscript satisfactory, and look forward to hearing from you.

Kind regards,

Gloriana Chaverri

REVIEWER: 1

Comments to the Author(s)

I commend the authors for submitting a very well written piece that is clear and nearly void of grammatical errors. There are only a few instances where clarification of the text is suggested (see the list of line items below). The introduction of the problem examined is well laid out and logical. And the discussion properly addresses the results and explores their significance even providing a good representation of caveats for the capture method they are espousing. The study is simple and straight forward, and I do not see any technical issues that need to be raised. Please see my list of suggestions below.

Authors’ response: We are very grateful to Dr. Angelo Soto-Centeno for his encouraging words.

Specific line points that could use a quick revision.

Abstract

Line 23: “Here we compare capture efficiency ...” – consider adding a comma after “Here, ...”

Authors’ response: We have modified this as suggested.

Introduction

Line 47: "... interpretation, which thus deteriorates ..." – delete the word "thus".

Authors' response: We have modified this as suggested.

Line 59: "A wide variety of methods exists ..." – delete the "s" in the word "exists".

Authors' response: We have modified this as suggested.

Line 61: "... methods for capturing and handling bats ..." – delete "and handling".

Authors' response: We have modified this as suggested.

Line 62: "On the other hand, indirect ..." – delete "On the other hand, ".

Authors' response: We have modified this as suggested.

Line 65: "... are considered more invasive ..." – delete "more ", it is not necessary as you did not include other methods in the sentence as comparison.

Authors' response: We have modified this as suggested.

Line 66: "... previous experience, especially during ..." – perhaps insert "by the user" to improve clarity.

Authors' response: We have modified this as suggested.

Line 67: "... ; however, they are essential ..." – use a period before "however" instead of semicolon to shorten the long running sentence.

Authors' response: We have modified this as suggested.

Line 71: "... detect and therefore avoid ..." – delete "therefore".

Authors' response: We have modified this as suggested.

Line 71–72: "... other trapping methods have been developed ..." – Some might argue that harp-traps were developed for capturing bats under different habitat circumstances instead of as a replacement for mist-nets. Truly, the sampling area of a harp-trap is much smaller than that of most mist-nets. So, these two cannot be directly compared. Maybe this fragment can be deleted and the sentence reworded a bit to avoid that direct comparison.

Authors' response: This is a good point. We have modified this section and have removed the 2 sentences that mention harp traps.

Results

Line 187: Figure 3 caption – I may have missed this... It seems like panels 0, 1, and 2 correspond to species richness, Shannon and Simpson diversity indices. It would make it more clear if this was added to the figure caption.

Authors' response: Good idea! We believe this makes the figure much easier to understand. We have thus included that information in the caption, which now reads: "Figure 3. Species accumulation curves (upper panels) and estimated sample coverage (lower panels) for the two types of nets used. The numbers above panels represent the Hill numbers, where $q = 0$ represents species richness, $q = 1$ represents the Shannon diversity index, and $q = 2$ represents the Simpson diversity index."

Discussion

Line 214: "... record low intensity calls." – insert the word "echolocation" after intensity to improve clarity.

Authors' response: We have modified this as suggested.

REVIEWER: 2

Comments to the Author(s)

Dear authors,

you present a field study comparing 2 types of mistnets that are used for catching bats. I only have 2 major points and some minor (see PDF):

1. Please clarify what you mean by traditional/regular mist nets. There are many types of mist nets that are used "traditionally" or "regularly" depending on the region of the world. Finding appropriate terms would be helpful.

Authors' response: This is a very good point; what is traditional or regular for some may not be so to others. We have included the term "Regular nylon or polyester mist nets" at the start of the abstract, and are explaining in the introduction and methods what we mean by this term (regular). Hopefully this solves the issue!

2. Please discuss a recently published work by Fereirra et al. 2021, Are bat mist nets ideal for capturing bats? From ultrathin to bird nets, a field test. Journal of Mammalogy. DOI: 10.1093/jmammal/gyab109.

Authors' response: Thank you for your recommendation. When we uploaded our manuscript, this paper had not been published yet, so we were unaware of its existence. We are now including this reference in several sections of our manuscript (abstract, last paragraph of the introduction, and third paragraph of the discussion).

Response to comments in pdf

Not sure what "traditional mist nets" are since this may vary across countries. Among bat scientist, there people that use "traditionally" monofilament nets, others use nylon mist nets ('hair-net') for decades. I think you should definitively clarify right at the beginning what you are talking about.

another term - 'regular mist net' - see comment above.

Authors' response: See response to first "major" comment above. Also, we have removed the word "traditional" and are keeping the word "regular" to maintain consistency of terms throughout the paper.

I think they are neither happy nor seek to get trapped. But based on their sensing ability, one species is easier to catch than another. And by the way, the reference is talking about personality (that's intraspecific!).

Authors' response: This is a good point. We have modified this sentence as follows, and have also changed the reference cited here: "A well-known phenomenon, for example, is the difference in trap-shy or trap-happy individuals, or those that are either less or more prone to being recaptured, respectively, which creates serious sampling biases [1]."

More invasive than ...? I am not sure if you can say this in general. Taking a bat from a hibernation site might be more invasive from physiological point of view.

Authors' response: We have removed the word "more" based on the suggestion of reviewer 1, which hopefully helps address the issue brought up by reviewer 2.

not necessary.

not necessarily. For individual health, probably.

Authors' response: We have modified this sentence according to both reviewers' suggestions: *"However, mist nets are essential for collecting information on morphometrics, to create acoustic libraries, understand sensitivity to ectoparasites, gauge species diversity, and estimate individual health, among others"*

and you also don't get much information... (e.g. sex, age, body condition, reproduction status, health, etc. and even number of individuals is missing)

Authors' response: This is a good suggestion. We have modified the sentence as follows: *"In addition, the use of acoustic recording equipment imposes higher economic costs than other existing capture methods, and does not provide data such as sex, age, body condition and other that may be relevant to a large number of studies."*

now I understand what is ment by 'traditional'. In my scene we call them bird nets as they originally have been created for birds. I know bat scientists that would call monofilament nets as traditional. Me, I 'grew up' with hair-nets. So, you see - it's difficult with the terms 'traditional' or 'regular' as they are not defined. Please try to find more accurate terms. hair nets are also nylon... but I guess this is not what you mean. e.g. see here different types sold by ecotone company: <https://en.ecotone.com.pl/mist-nets-2,3,93/r0/SGFpci1uZXQ=.html> Hair-nets are also not included. You should still mention those because at in Europe, they are commonly used!

Authors' response: We have modified this entire section to incorporate the suggestions of reviewer 2. Now it reads as follows: *"Some of the types of mist nets that have been developed, according to their netting, include polyester and nylon which have thicker threading, and ultrathin nets such as hair and monofilament nets. The latter types were developed as a novel tool for bat trapping, and these could have great potential to minimize some of the detectability limitations of other types of mist nets, while still benefiting from the already well-described advantages of this trapping method. Monofilament nets are designed to be less detectable by bats, as their netting is made of single strands of yarn, unlike regular (i.e., polyester and nylon) mist nets whose netting is created by twisting several individual strands, resulting in a thicker netting material."*

I am not sure what this means. You added that from the literature into the analysis?

Authors' response: We have modified this sentence and hope the new wording provides a clearer explanation. *"Finally, we established the diet of each species sampled based on information gathered from previous studies [26]."*

I think you should definitively cite and discuss this paper at some point since it's similar to your work: Diogo F Ferreira, Crinan Jarrett, Patrick Jules Atagana, Luke L Powell, Hugo Rebelo, Are bat mist nets ideal for capturing bats? From ultrathin to bird nets, a field test, Journal of Mammalogy, 2021;, gyab109, <https://doi.org/10.1093/jmammal/gyab109>

Authors' response: See response to previous comments on the topic.

Here, I do not agree. E.g. hair-nets also have multiple strands, however these are very thin. In my experience, hair-nets are similar to monofilament or even better (however this is more a gut feeling rather than I have appropriate data). These net type is missing in your study and therefore it cannot be generalized. Nevertheless I understand that this was not possible because in the tropics this sensitive net type might be single-use for your large bats. :-)

Authors' response: We have modified the wording: *"Thus, net design, particularly with regards to the thickness of the strands used for constructing the netting, is an important factor to consider when the main interest of the study is to gauge species diversity or when certain trophic guilds, especially insectivorous, are targeted in bat surveys."*

unclear what is meant here... I find it also difficult to compare birds and bats since they have a completely different sensing (visual versus ultrasound).

Authors' response: we have removed this section based on the reviewer's suggestion.

Well, I wouldn't be too much afraid of the net breaking... the injuries are a more important argument.

Authors' response: We agree that the injuries are the more important issue here, yet for scientists with few resources, net damage is also something important to consider. However, we have modified this section and are now mostly focusing on comparing our results to those of Ferreira et al., so hopefully we have addressed the issue raised here by the reviewer.